# The Pitfalls of Heterosis Coefficients

**DOI:** 10.3390/plants9070875

**Published:** 2020-07-09

**Authors:** Dominique de Vienne, Julie B. Fiévet

**Affiliations:** GQE—Le Moulon, INRAE, Université Paris-Saclay, CNRS, AgroParisTech, 91190 Gif-sur-Yvette, France; julie.fievet@inrae.fr

**Keywords:** flowering, growth rate, heterosis measurement, hybrid vigour, maize, non-additivity, potence ratio, yield

## Abstract

Heterosis (hybrid vigour) is a universal phenomenon of crucial agro-economic and evolutionary importance. We show that the most common heterosis coefficients do not properly measure deviation from additivity because they include both a component accounting for “real” heterosis and a term that is not related to heterosis, since it is derived solely from parental values. Therefore, these coefficients are inadequate whenever the aim of the study is to compare heterosis levels between different traits, environments, genetic backgrounds, or developmental stages, as these factors may affect not only the level of non-additivity, but also parental values. The only relevant coefficient for such comparisons is the so-called “potence ratio”. Because most heterosis studies consider several traits/stages/environmental conditions, our observations support the use of the potence ratio, at least in non-agronomic contexts, because it is the only non-ambiguous heterosis coefficient.

## 1. Introduction

Non-linear processes are extremely common in biology. In particular, genotype-phenotype or phenotype-phenotype relationships frequently display concave behaviours, resulting in the dominance of “high” over “low” alleles [1] and in positive heterosis for a wide range of polygenic traits [2,3]. Properly quantifying the degree of non-additivity is an essential prerequisite for interpreting and comparing genetic studies and for making predictions in plant and animal breeding. In this commentary paper, we first recap the different ways non-additivity is measured in genetics. Subsequently, we analyse the formal relationships between the different heterosis coefficients and provide examples drawn from experimental studies in maize and cotton. Finally, we show the extent to which the most commonly used heterosis coefficients may lead to interpretation errors.

## 2. The Dominance and Heterosis Coefficients

There are two classical coefficients to measure the degree of dominance:(i)Wright [1] defined:
$$D_{W}=\frac{z_{2}-z_{12}}{z_{2}-z_{1}}$$
where $z_{1}$, $z_{2}$ and $z_{12}$ are, respectively, the phenotypic values of genotypes $A_{1}A_{1}$, $A_{2}A_{2}$ and $A_{1}A_{2}$, with $z_{2}>z_{1}$. $D_{W}$ varies from 0, when $A_{2}$ is fully dominant over $A_{1}$, to 1, when $A_{2}$ is fully recessive with respect to $A_{1}$. $D_{W}=0.5$ corresponds to semi-dominance or additivity ($z_{12}=\frac{z_{1}+z_{2}}{2}$) (Table 1). Note that $D_{W}$ is strictly equivalent to the dominance coefficient *h* used in evolutionary genetics [4].(ii)Falconer [5] proposed the following coefficient:
$$D_{F}=\frac{z_{12}-\bar{z}}{\frac{z_{2}-z_{1}}{2}}$$
where $\bar{z}=\frac{z_{1}+z_{2}}{2}$. $D_{F}$ varies in the opposite direction to $D_{W}$: its value is 1 if $z_{12}=z_{2}$ (complete dominance of $A_{2}$ over $A_{1}$), −1 if $z_{12}=z_{1}$ ($A_{2}$ is fully recessive with respect to $A_{1}$) and 0 if there is additivity. In the case of overdominance, $D_{W}<0$ and $D_{F}>1$, and in the case of underdominance, $D_{W}>1$ and $D_{F}<-1$ (Table 1).

The $D_{W}$ and $D_{F}$ coefficients are linearly related:$$D_{F}=1-2D_{W}$$

Thus, dominance can be quantified with either coefficient, since both of them give the position of the heterozygote relative to the parental homozygotes.

For polygenic traits, either coefficient could be used to quantify non-additivity, i.e., “real” heterosis, without any ambiguity. Actually, one finds five heterosis coefficients in the literature (see their characteristic values in Table 1).

The two most popular coefficients are the best-parent (BP) and mid-parent (MP) heterosis coefficients (e.g., [6,7]):$$H_{\mathrm{BP}}=\frac{z_{12}-z_{2}}{z_{2}}$$
$$H_{\mathrm{MP}}=\frac{z_{12}-\bar{z}}{\bar{z}}$$
where $z_{2}$, $z_{12}$ and $\bar{z}$ are, respectively, the phenotypic values of the parent 2 (with $z_{2}>z_{1}$), of the parent 1 × parent 2 hybrid and of the parental mean.

In some instances, the authors do not normalize the difference between the hybrid and the best- or mid-parent value. Fonseca & Patterson [8] proposed:$$H_{\mathrm{bp}}=z_{12}-z_{2}$$
and Falconer [5]:$$H_{\mathrm{mp}}=z_{12}-\bar{z}$$

Finally, the so-called “potence ratio” [9] has the same expression as Falconer’s dominance coefficient ($D_{F}$):$$H_{\mathrm{PR}}=\frac{z_{12}-\bar{z}}{\frac{z_{2}-z_{1}}{2}}.$$

A value of 0 indicates additivity, 1 indicates $z_{12}=z_{2}$ (hybrid value = best-parent value), −1 indicates $z_{12}=z_{1}$ (hybrid value = worst-parent value), and >1 (resp. <−1) indicates best-parent (resp. worst-parent) heterosis (Table 1). $H_{\mathrm{PR}}$*explicitly* includes the values of the three genotypes, whereas the other coefficients lack one of the parental values ($H_{\mathrm{BP}}$ and $H_{\mathrm{bp}}$) or both ($H_{\mathrm{MP}}$ and $H_{\mathrm{mp}}$—a given mean can correspond to an infinity of parental values). From a genetic point of view, $H_{\mathrm{PR}}$ is explicitly expressed in terms of the five genetic effects contributing to heterosis (Appendix A). Thus, the potence ratio, which is still by far the least used heterosis coefficient, is the only one that informs us of the exact position of the hybrid value relative to the parental values. Wright’s dominance coefficient has the same property, but its inverse direction of variation, which makes comparisons less easy, probably explains why it is not used in this context.

## 3. Relationship between the Potence Ratio and the other Heterosis Coefficients

It is easy to show that the relationship between $H_{\mathrm{PR}}$ and the other coefficients is (with $z_{2}>z_{1}$):(1)$$\begin{array}{cc} \; & H_{\mathrm{MP}}=H_{\mathrm{PR}}z_{m} \end{array}$$
(2)$$\begin{array}{cc} \; & H_{\mathrm{BP}}=\left(\frac{-1+H_{\mathrm{PR}}}{2}\right)z_{b} \end{array}$$
(3)$$\begin{array}{cc} \; & H_{\mathrm{mp}}=H_{\mathrm{PR}}\bar{z}z_{m} \end{array}$$
(4)$$\begin{array}{cc} \; & H_{\mathrm{bp}}=\left(\frac{-1+H_{\mathrm{PR}}}{2}\right)z_{2}z_{b} \end{array}$$
where $z_{m}=\frac{z_{2}-z_{1}}{z_{1}+z_{2}}$ (the difference between the parental values normalized by their sum) and $z_{b}=\frac{z_{2}-z_{1}}{z_{2}}$ (the difference between the parental values normalized by the best parental value).

For a given $H_{\mathrm{PR}}$ value, the coefficients $H_{\mathrm{MP}}$ and $H_{\mathrm{BP}}$ are linearly related to $z_{m}$ and $z_{b}$, respectively, i.e., they depend on the scale of the parental values. More specifically, the relationship between $H_{\mathrm{MP}}$ and $z_{m}$ is negative when $H_{\mathrm{PR}}<0$ and positive when $H_{\mathrm{PR}}>0$, while the relationship between $H_{\mathrm{BP}}$ and $z_{b}$ is negative when $H_{\mathrm{PR}}<1$ and positive when $H_{\mathrm{PR}}>1$. As $z_{m}$ and $z_{b}$ are positive, we see from Equation (Equation 1) that for $H_{\mathrm{PR}}\neq 0$, we have
$$0<H_{\mathrm{MP}}<H_{\mathrm{PR}}\;\;\;\;\mathrm{if}\;H_{\mathrm{PR}}>0\;\;\left(\mathrm{positive}\;\mathrm{mid}\text{-}\mathrm{parent}\;\mathrm{heterosis}\right)$$
and
$$H_{\mathrm{PR}}<H_{\mathrm{MP}}<0\;\;\;\;\mathrm{if}\;H_{\mathrm{PR}}<0\;\;\left(\mathrm{negative}\;\mathrm{mid}\text{-}\mathrm{parent}\;\mathrm{heterosis}\right)$$

Regarding $H_{\mathrm{BP}}$, we see from Equation (Equation 2) that, for $H_{\mathrm{PR}}\neq 1$, we have
$$0<H_{\mathrm{BP}}<\left(\frac{-1+H_{\mathrm{PR}}}{2}\right)\;\;\;\;\mathrm{if}\;H_{\mathrm{PR}}>1\;\;\left(\mathrm{positive}\;\mathrm{best}\text{-}\mathrm{parent}\;\mathrm{heterosis}\right)$$
and
$$\left(\frac{-1+H_{\mathrm{PR}}}{2}\right)<H_{\mathrm{BP}}<0\;\;\;\;\mathrm{if}\;H_{\mathrm{PR}}<1\;\;\left(\mathrm{negative}\;\mathrm{best}\text{-}\mathrm{parent}\;\mathrm{heterosis}\right)$$

If $H_{\mathrm{PR}}=0$ (resp. $H_{\mathrm{PR}}=1$), $H_{\mathrm{MP}}$ (resp. $H_{\mathrm{BP}})=0$.

Numerical applications performed with nine $H_{\mathrm{PR}}$ values, from $H_{\mathrm{PR}}=-2$ to $H_{\mathrm{PR}}=2$, show that the same $H_{\mathrm{MP}}$ or $H_{\mathrm{BP}}$ value can be observed for very different $H_{\mathrm{PR}}$ values, depending on $z_{m}$ and $z_{b}$ values, respectively (Appendix A). For instance, $H_{\mathrm{MP}}\approx 0.4$ can correspond to both mid-parent heterosis ($H_{\mathrm{PR}}=0.5$, $z_{m}\approx 0.8$) and best-parent heterosis ($H_{\mathrm{PR}}=2$, $z_{m}\approx 0.21$).

We illustrate this by using experimental data from maize. We measured six traits (flowering time, plant height, ear height, grain yield, thousand-kernel weight, and kernel moisture) in four crosses (B73 × F252, F2 × EP1, F252 × EP1, F2 × F252) grown in three different environments in France (Saint-Martin-de-Hinx in 2014, Jargeau in 2015, and Rhodon in 2015). We computed $H_{\mathrm{PR}}$, $H_{\mathrm{MP}}$ and $H_{\mathrm{BP}}$ for the 72 trait-cross-environment combinations. Figure 1A,B shows that the relationship between $H_{\mathrm{PR}}$ and the other two coefficients is very loose, if any. A given $H_{\mathrm{PR}}$ value can correspond to a wide range of $H_{\mathrm{MP}}$ or $H_{\mathrm{BP}}$ values, and vice versa. We performed the same analyses using the data published by Shang et al. [10], who measured five traits in two crosses of cotton grown in three environments. The same loose relationship between $H_{\mathrm{PR}}$ and either heterosis coefficient was observed (Figure 1C,D). This means that the normalized differences between the parents, which are not related to heterosis, since they do not include values from the hybrids, markedly affect $H_{\mathrm{MP}}$ and $H_{\mathrm{BP}}$.

Regarding the $H_{\mathrm{mp}}$ and $H_{\mathrm{bp}}$ coefficients, which are not dimensionless, they only provide the direction of heterosis. For a given $H_{\mathrm{PR}}$ value, $H_{\mathrm{mp}}$ can vary from −∞ to 0 when $H_{\mathrm{PR}}<0$ and from 0 to +∞ when $H_{\mathrm{PR}}>0$, and $H_{\mathrm{bp}}$ can vary from −∞ to 0 when $H_{\mathrm{PR}}<1$ and from 0 to +∞ when $H_{\mathrm{PR}}>1$ (Equations (Equation 3) and (Equation 4)).

Let us examine the possible interpretation errors that may result from the use of $H_{\mathrm{MP}}$ and $H_{\mathrm{BP}}$.

## 4. The Pitfalls of the Most Commonly Used Heterosis Coefficients

The non-univocal relationship between $H_{\mathrm{PR}}$ and the most commonly used heterosis coefficients has two consequences. (i) Comparing the coefficient values for a given trait in different crosses and/or environments and/or developmental stages leads to erroneous conclusions whenever these factors have an effect on the scale of the trait and/or on the difference between the parental values (i.e., on $z_{m}$ or $z_{b}$). Possible differences in deviations from additivity between these conditions cannot be detected. (ii) This problem is even more pronounced when studying different traits, because each trait has its own scale of variation, making $H_{\mathrm{MP}}$ and $H_{\mathrm{BP}}$ (and to a greater extent $H_{\mathrm{mp}}$ and $H_{\mathrm{bp}}$) useless for comparing the real levels of heterosis of these traits.

These pitfalls can easily be illustrated from our maize dataset. Figure 2A shows that classifying a set of traits according to their degree of heterosis can give markedly different results, depending on whether one uses the $H_{\mathrm{PR}}$ coefficient or one of the two coefficients $H_{\mathrm{MP}}$ and $H_{\mathrm{BP}}$. For instance, in the F252 × EP1 cross, flowering time displays moderate heterosis according to $H_{\mathrm{MP}}$ and $H_{\mathrm{BP}}$ even though this trait actually has the highest $H_{\mathrm{PR}}$ value. Conversely, plant height is the second most heterotic trait according to $H_{\mathrm{MP}}$ and $H_{\mathrm{BP}}$, but not according to $H_{\mathrm{PR}}$. Similarly, comparing heterosis of a given trait in different hybrids results in coefficient-specific rankings: heterosis of ear height measured with $H_{\mathrm{PR}}$ is highest in the B73 × F252 hybrid, whereas according to $H_{\mathrm{MP}}$ and $H_{\mathrm{BP}}$ the highest values are found in the F252 × EP1 hybrid (Figure 2B). Finally, the effect of the environment on heterosis also reveals obvious discrepancies between $H_{\mathrm{PR}}$ on the one hand and $H_{\mathrm{MP}}$ or $H_{\mathrm{BP}}$ on the other hand (Figure 2C).

It is also informative to compare the profiles of heterosis coefficients for a trait measured during development or growth. A Hill function was used to fit the percentage of flowering individuals over time in the W117 × F192 and W117 × F252 hybrids and their parents:$$y=\frac{ax^{n}}{b+x^{n}}$$
where *x* is the time, *a* and *b* are constants, and *n* is the Hill coefficient. We then computed the heterosis coefficients over time for the percentage of flowering individuals estimated from the fitted curves (Figure 3). Again, $H_{\mathrm{PR}}$ tells a different story when compared to $H_{\mathrm{MP}}$ and $H_{\mathrm{BP}}$. Because hybrid and parental values converge as flowering progresses, both $H_{\mathrm{MP}}$ and $H_{\mathrm{BP}}$ inevitably decrease when flowering nears 100%. The evolution of $H_{\mathrm{PR}}$, which quantifies the "real" heterosis, is clearly different, with a monotonous increase in the W117 × F192 hybrid and a fluctuating profile in the W117 × F252 hybrid.

Similar results were observed in a simulation describing the increase in population size of a unicellular organism, which exhibits logistic growth. We used:$$y=\frac{K}{1+ae^{-r\theta }}$$
where *y* is the size of the population, *K* the carrying capacity, *a* a constant, *r* the growth rate, and θ the time. We assumed that the parents only differed in their growth rate *r* and that there was additivity for this parameter. The results show that the $H_{\mathrm{MP}}$ and $H_{\mathrm{BP}}$ profiles for population size over time are clearly not congruent with that of $H_{\mathrm{PR}}$ (Appendix A).

## 5. Discussion

Using simple theoretical considerations and relying on experimental data and simulations, we showed that the most commonly used heterosis coefficients, i.e., $H_{\mathrm{MP}}$ and $H_{\mathrm{BP}}$ (and their non-normalized forms $H_{\mathrm{mp}}$ and $H_{\mathrm{bp}}$) cannot and should not be used if the heterosis levels are to be compared between different traits, environments, genetic backgrounds, or developmental stages. Because their expression does not explicitly include the two parental values in addition to the hybrid value, these coefficients, unlike the potence ratio $H_{\mathrm{PR}}$, do not quantify the deviation from additivity but only the normalized distance between the hybrid value and either the best or the mean parental value. The extent to which erroneous conclusions can be drawn when performing comparisons using these coefficients was illustrated with data from maize and cotton, and from population growth simulation in a micro-organism.

If $H_{\mathrm{MP}}$, $H_{\mathrm{BP}}$, $H_{\mathrm{mp}}$, and $H_{\mathrm{bp}}$ do not provide reliable information on non-additivity, why are they so commonly used? There are probably both historical and technical reasons: (i) the first scientists who quantified heterosis were plant breeders [11,12]. From an economic perspective, the goal was, and still is, to develop hybrids that are “better” than the best- or mid-parent values for the desired agronomic traits, and not to know where the hybrid value is relative to the parental values. Heterosis coefficients have been defined accordingly and the habit has remained; (ii) the coefficients giving the right non-additivity values, $H_{\mathrm{PR}}$ for heterosis and $D_{W}$ or $D_{F}$ for dominance, can take on high to very high values when the parents are close, due to the small differences $z_{2}-z_{1}$ in the denominator of the fractions. This can produce extreme values that are not easy to represent and manipulate for statistical treatments. Nevertheless, such values are biological realities that precisely convey the inheritance of the traits under study, something that $H_{\mathrm{MP}}$, $H_{\mathrm{BP}}$, $H_{\mathrm{mp}}$, and $H_{\mathrm{bp}}$ do not. Note that the two dominance coefficients used for monogenic traits have the same property, which does not prevent their use to the exclusion of any other. In addition, from a practical point of view, a single coefficient is sufficient to know the position of the hybrid relative to the mid- or best-parent, whereas in a number of studies the authors compute and comment both $H_{\mathrm{MP}}$ and $H_{\mathrm{BP}}$ (or $H_{\mathrm{mp}}$ and $H_{\mathrm{bp}}$). In conclusion, to compare the amplitude of heterosis between traits, developmental stages, crosses, or environmental conditions, there is no other choice but to use the only heterosis coefficient— $H_{\mathrm{PR}}$ —that is not affected by the scale of the parental values and that accounts for the position of the hybrid in the parental range.

## Figures and Tables

**Figure 1 plants-09-00875-f001:**
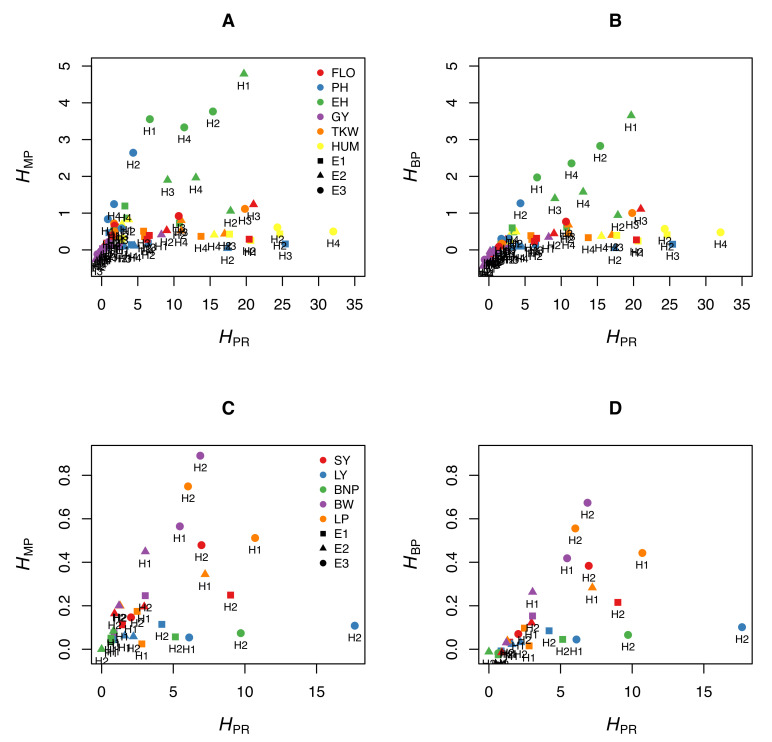
Relationship between the potence ratio $H_{\mathrm{PR}}$ and the two heterosis coefficients $H_{\mathrm{MP}}$ and $H_{\mathrm{BP}}$. (**A**,**B**) Six traits were measured in maize (FLO: flowering time [days between 50% flowering and 12 August], PH: plant height, EH: ear height, GY: grain yield, TKW: thousand-kernel weight, KM: kernel moisture) in four crosses (H1: B73 × F252, H2: F2 × EP1, H3: F252 × EP1, H4: F2 × F252) grown in three environments in France (E1: Saint-Martin-de-Hinx, E2: Jargeau, E3: Rhodon). (**A**) Relationship between $H_{\mathrm{PR}}$ and $H_{\mathrm{MP}}$. (**B**) Relationship between $H_{\mathrm{PR}}$ and $H_{\mathrm{BP}}$. (For clarity, four of the 72 trait-cross-environment combinations are not represented because they have high $H_{\mathrm{PR}}$ values.) (**C**,**D**) Five traits were measured in cotton (SY: seed yield [grams per plant], LY: lint yield [grams per plant], BNP: bolls per plant, BW: boll weight [grams], LP: lint percentage) in two crosses (H1: X1135 × GX100-2 and H2: GX1135 × VGX100-2) grown in three environments in China (E1: Handan, E2: Cangzhou, E3: Xiangyang) (Data from [10]). (**C**) Relationship between $H_{\mathrm{PR}}$ and $H_{\mathrm{MP}}$. (**D**) Relationship between $H_{\mathrm{PR}}$ and $H_{\mathrm{BP}}$.

**Figure 2 plants-09-00875-f002:**
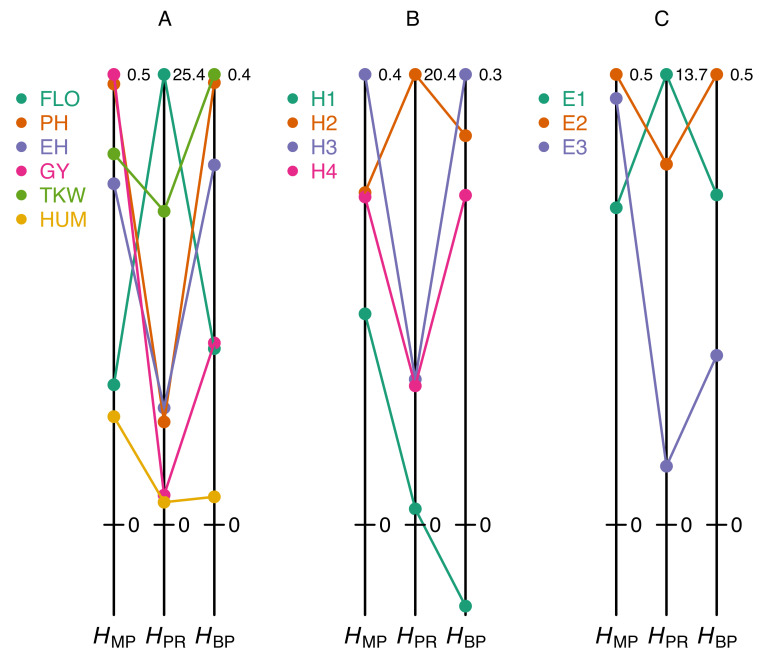
Heterosis values obtained with different coefficients. (**A**) Heterosis coefficients for six traits measured in the F252 × EP1 cross grown in Saint-Martin-de-Hinx (France) in 2014. (**B**) Heterosis coefficients for ear height in four crosses grown in Saint-Martin-de-Hinx (France) in 2014. (**C**) Heterosis coefficients for plant height in the F2 × F252 cross grown in the three environments. The six traits and the three environments are the same as in Figure 1A. The scales of the heterosis coefficients are normalized by the maximum value in each dataset (figures at the top right of the vertical lines).

**Figure 3 plants-09-00875-f003:**
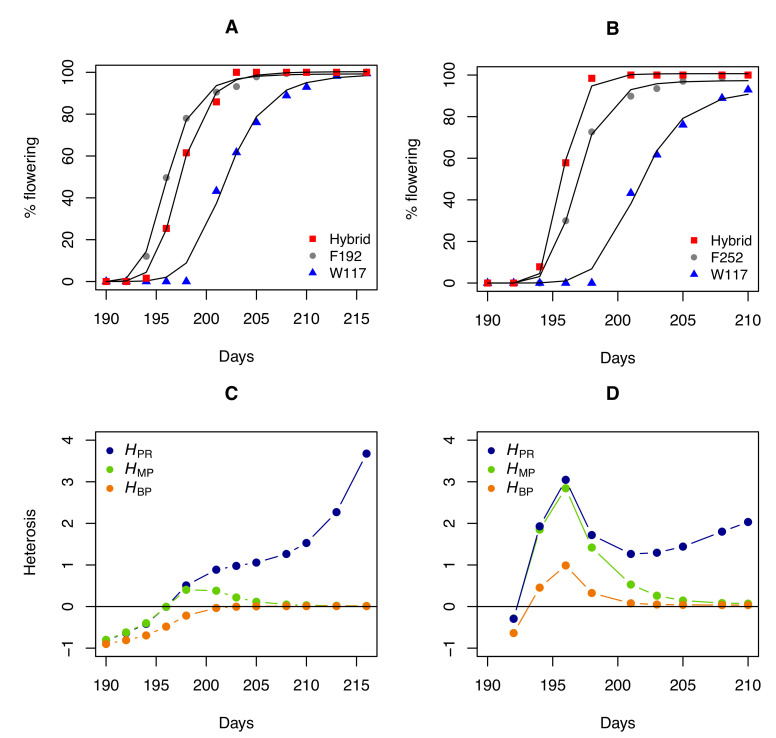
Heterosis for flowering in two maize hybrids. (**A**) Percentage of flowering over time (number of days since 1 January) for parents W117 and F192 and their hybrid, adjusted with a Hill function. (**B**) Percentage of flowering over time for parents W117 and F252 and their hybrid. (**C**) Heterosis coefficient profiles over time for the W117 × F192 cross. (**D**) Heterosis coefficient profiles over time for the W117 × F252 cross.

**Table 1 plants-09-00875-t001:** Dominance and heterosis coefficients. $D_{W}$: Wright’s dominance coefficient [1]. $D_{F}$: Falconer’s dominance coefficient [5]. $H_{\mathrm{mp}}$, $H_{\mathrm{MP}}$, $H_{\mathrm{PR}}$, $H_{\mathrm{bp}}$ and $H_{\mathrm{BP}}$: heterosis coefficients. Subscripts: mp or MP, mid-parent; PR, potence ratio; bp or BP, best-parent. $z_{1}$ (resp. $z_{2}$): the phenotypic value of parental homozygote 1 or of parent 1 (resp. 2). $z_{12}$: the heterozygote or hybrid value. $\bar{z}$: the mean parental value. By convention, $z_{2}>z_{1}$.

Reference	Coefficient	Coefficient Scales with Their Characteristic Values
Highhomozygote	DW=z2−z12z2−z1	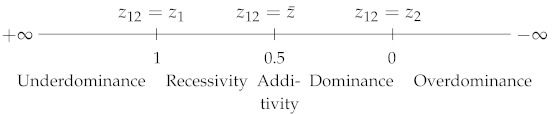
Meanhomozygote	DF=z12−z¯(z2−z1)/2	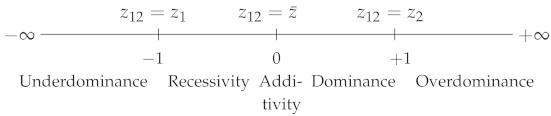
Mid-parent	Hmp=z12−z¯	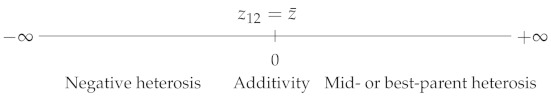
HMP=z12−z¯z¯	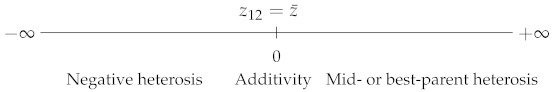
HPR=z12−z¯(z2−z1)/2	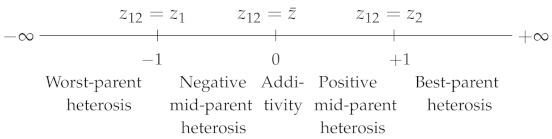
Best-parent	Hbp=z12−z2	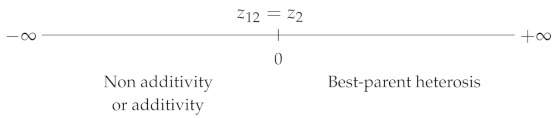
HBP=z12−z2z2	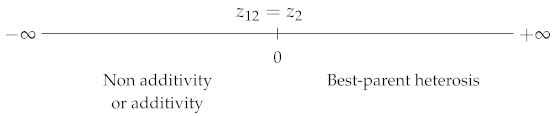

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
