# Peer review of "The Pitfalls of Heterosis Coefficients"

_plants, 2020, doi:10.3390/plants9070875_

Round 1

Reviewer 1 Report

Title: The pitfalls of heterosis coefficients

The subject of this manuscript falls within the general scope of Plants. The manuscript is an original contribution to the indexes recording heterosis of hybrids. The manuscript need to be rearranged to show more clearly this interesting study. Then, this manuscript should be reconsider after major revision

Keywords should be in alphabetical order. Avoid words included in the title.

Introduction

Lines 42-43. Delete ‘However, most of the classical heterosis coefficients do not meet  this requirement?. I do not think you should advance your results here.

Lines 62-63 Delete ‘The two coefficients merely vary in opposite directions, with DW

decreasing when dominance increases’. This sentence repeats previous information.

Lines 73-74. Please add a reference, example of a study, for this sentence.

Lines 86-91. I think you should not advance your result about validation of the potence ratio in the Introduction.

In general, I think you should rearrange Introduction and the two results section before Discussion to firstly introduce the problem you are dealing with, then explain what your analysis in a material and methods section and then show your results (what you do now in the Introduction and the next two sections).

Reviewer 2 Report

This paper mainly suggests to divide measures of heterosis by the difference between two parental means. The reason for the suggestion is mainly based on the one-locus argument and interpretation. The question is: Is this helpful or not? The short answer is: It is not helpful at all, and could be problematical in interpretation in the context of multiple loci.

The discussion in page 3-4 (and other places) is implicitly based on the one-locus interpretation or logic (specifically for Table 1). Of course, many quantitative traits are affected by multiple loci, as acknowledged in supplemental information. When dealing with multiple loci, one needs to recognize that not all plus alleles are fixed in high line and not all minus alleles are fixed in low line. In this case, the parental difference or range is not that simple and straightforward for the genetic interpretation, not what you have in mind in leading the discussion in page 3-4 and in Table 1.

A very important example is the heterosis in maize yield. In a classic cross for the study of maize heterosis between B73 and MO17, there is not much difference in yield between parents (due to inbreeding depression, the opposite phenomenon of heterosis) and huge difference between hybrid and parent means (Stuber et al. 1992 Genetics). This is because the QTL additive effects are about equally positive or negative between loci, and the QTL dominance effects are all positive (Garcia et al. 2008 Genetics). The heterosis as measured by Hmp is due to QTL dominance and additive x additive interaction (based on a F2 genetic model, more discussion on this below). There is overwhelm evidence that in this case the heterosis is primarily due to dominance. Thus by using QTL analysis, we can reach a pretty good understanding on the cause and level of heterosis. I do not think it is helpful at all to divide Hmp by the parental range, as the parental range (which should just be the sum of QTL additive effects) can vary quite differently and has little bearing on Hmp. Hpr not only has a complicated genetic interpretation unnecessarily (supplemental information), and it does not shed much light for our understanding at all from a multiple locus point of view (the single locus interpretation as in Table 1 does not help at all for quantitative traits).

The paper is loaded with naïve thinking and wording, such as “actual heterosis”, “real level of heterosis”, “erroneous conclusions”, “biological realties”, …. It is one thing to use a quantity to help our understanding if it does help. (In this case it actually does not, as pointed out above). But it would be quite different (actually naivete and downright silly) to regard it as “real” and “actual” and others as “erroneous”. The tone of discussion in page 9 is very silly and loaded with unsubstantiated words. This should not appear in any mature publication.

Another point: The authors are fully aware that they use a F-infinity model to interpret heterosis. As pointed out in Zeng et al (2008 Genetics), F-infinity model has a problem on the consistency. It is particularly problematic for interpreting heterosis.  

Reviewer 3 Report

The study describes the pitfalls of interpreting heterosis coefficients by comparing five coefficients found in literature.

The objectives/pitfalls of the study/heterosis coefficients was clearly stated in the study. Overall, the study is clearly structured and well written. Only minor clarifications are necessary.

Figure 2 shows the comparison of heterosis levels calculated by three different coefficients. For clarity it is recommended to use the same scale for the three coefficients. In addition, Figure 2C displays the comparison of the heterosis coefficients between three different environments. The authors state that the effect of environment on heterosis reveal the same discrepancies as seen in A and B of the different coefficients. However, the level of heterosis for a specific trait is environmental dependent so that it is difficult to compare different coefficients between different environments.

In line 165 the Hill function was used to fit the percentage of flowering over time and it was mentioned that n represents the Hill coefficients. However there is no explanation for the remaining model terms a, b and x, which needs to be included.

Figure 3 represents heterosis and estimated heterosis coefficients for flowering time in maize. Based on these results the authors stated that the coefficients HMP and HBP do tell a different story than the potence ratio:" Both HMP and HBP decrease as flowering progresses because the normalized di fference between the parents also decreases. This masks the evolution of real heterosis, which actually increases." The trait "flowering" might not be the right choice for this conclusion, as the hybrids reached ~100% flowering (Figure A and B) at day ~200/195 which is confirmed with the peak of the heterosis coefficients in Figure C and D. After the hybrids reached 100% flowering the level of heterosis should increase as also the parents get close to their maximum in flowering.

In line 188, the authors write that HMP does not quantify the deviation form additivity. However, the HMP describe the deviation of the hybrid from the mean of the parents (Mid parental value) and hence does describe the difference from additivity (displayed in Table 1).

Figure 1 needs some modification. The labels of the data points with in the plots are partly unreadable due to overlaying and/or cropped labels. In addition, the units of the phenotypic traits are missing in the legend.

hP was introduced for simplicity instead of HPR, however this is not necessary - HPR is more plausible.

Round 2

Reviewer 1 Report

The authors have answered correctly to my previous comments.

Reviewer 2 Report

Nothing is changed. The previous evaluation still remains.